# Clinical Presentation and Prognostic Features in Patients with Immunotherapy-Induced Vitiligo-like Depigmentation: A Monocentric Prospective Observational Study

**DOI:** 10.3390/cancers14194576

**Published:** 2022-09-21

**Authors:** Nicola Hermann, Lara Valeska Maul, Milad Ameri, Stephan Traidl, Reihane Ziadlou, Karolina Papageorgiou, Isabel Kolm, Mitchell Levesque, Julia-Tatjana Maul, Marie-Charlotte Brüggen

**Affiliations:** 1Department of Dermatology, University Hospital of Zurich, 8091 Zurich, Switzerland; 2Faculty of Medicine, University of Zurich, 8006 Zurich, Switzerland; 3Department of Dermatology, University Hospital of Basel, 4031 Basel, Switzerland; 4Medical Campus Davos, 7265 Davos, Switzerland; 5Department of Dermatology and Allergy, Hannover Medical School, 30625 Hannover, Germany

**Keywords:** melanoma, checkpoint inhibitors, immune-related toxicity, vitiligo, survival, LDH, BRAF, vitiligo-like depigmentation

## Abstract

**Simple Summary:**

In this study, we thoroughly explore the clinical and biological features of immunotherapy-induced vitiligo-like depigmentation (VLD) in patients with stage III-IV melanoma receiving immunotherapy. Key findings include a distinct immune signature with an upregulation of ITGA in the VLD group, an upregulation of EDAR and downregulation of LAG3 in VLD-responders group, a longer time to VLD onset in patients pre-treated with targeted therapy, the distribution of VLD lesions primarily in sun-exposed areas as opposed to mechanically stressed areas in non-immunotherapy-related vitiligo, the predominance of a symmetrical, small-freckle lesional pattern and a prolonged progression-free survival (PFS) and overall survival (OS) in VLD patients without LDH elevation as opposed to those with LDH elevation.

**Abstract:**

Vitiligo-like depigmentation (VLD) is an immune-related adverse event (irAE) of checkpoint-inhibitor (CPI) treatment, which has previously been associated with a favourable outcome. The aim of this study was to explore clinical, biological and prognostic features of melanoma patients with VLD under CPI-treatment and to explore whether they exhibit a characteristic immune response profile in peripheral blood. Melanoma patients developing VLD under CPI were included in a prospective observational single-center cohort study. We collected and analysed clinical parameters, photographs and serum from 28 VLD patients. They received pembrolizumab (36%), nivolumab (11%), ipilimumab/nivolumab (32%) or clinical trial medications (21%). We performed a high-throughput proteomics assay (Olink), in which we identified a distinct proteomic signature in VLD patients in comparison to non-VLD CPI patients. Our clinical assessments revealed that VLD lesions had a predominantly symmetrical distribution pattern, with mostly smaller “freckle-like” macules and a preferential distribution in UV-exposed areas. Patients with previous targeted therapy showed a significantly longer time lapse between CPI initiation and VLD onset compared to non-pre-treated patients (12.5 vs. 6.25 months). Therapy responders exhibited a distinct proteomic profile when compared with non-responders in VLD such as upregulation of EDAR and downregulation of LAG3. ITGA11 was elevated in the VLD-group when compared to non-VLD-CPI-treated melanoma patients. Our findings demonstrate that on a proteomic level, VLD is characterized by a distinct immune signature when compared to CPI-treated patients without VLD and that therapy responsiveness is reflected by a characteristic immune profile. The pathomechanisms underlying these findings and how they could relate to the antitumoral response in melanoma remain to be elucidated.

## 1. Introduction

Checkpoint inhibitors (CPI) and targeted therapies have revolutionized the treatment for metastatic melanoma over the past decade [1,2,3,4]. The most commonly used CPIs are monoclonal antibodies directed against cytotoxic T lymphocyte antigen-4 (CTLA-4) and programmed cell death-1 (PD-1). Both have demonstrated a remarkable improvement in the overall survival in advanced melanoma [1].

CTLA-4 and PD1 are both regulators of the T-cell response [5]. CTLA-4 dampens T lymphocyte activation in an early stage, while PD1-mediated immune regulation takes place in the effector phase. Antibodies directed against CTLA-4 and PD-1 unleash the T-cell response against the tumour, but can also trigger a series of immune-related adverse events (irAEs). While irAEs seem to be a sign of therapy responsiveness [6], they can be very severe [1] or even fatal. Balancing between antitumoral response and irAEs remains one of the major challenges in the current melanoma treatment landscape. The combination of anti-CTLA4 and anti-PD1 causes irAEs of any grade in 96% of the patients and of grade 3–4 in up to 59% of the patients [1,7,8]. The skin is the most commonly affected target organ of irAEs [7,8]. Cutaneous irAEs range from mild self-limiting rashes, eczema, psoriasis-like or lichen planus-like dermatitis to alopecia areata, and, most interestingly, vitiligo-like depigmentation (VLD). Current data suggest that the occurrence of cutaneous irAEs, especially rashes and vitiligo, may be associated with better treatment outcomes [9,10,11]. In a prospective study with 67 melanoma patients under pembrolizumab, complete or partial response to treatment was more frequent in patients who developed VLD [10]. In a retrospective analysis of nivolumab-treated melanoma patients, the overall survival (OS) was significantly longer in patients who developed VLD [12]. 

VLD occurs in 4.7–37.5% of the melanoma patients undergoing anti-PD1 treatment and in 2.9–4.9% of the melanoma patients treated with CTLA-4-antibodies [13,14,15,16]. VLD manifests with depigmented patches, mostly in UV-exposed skin [17,18]. 

There have been attempts to characterize VLD on a cytokine level [19,20]. However, little is known about the distinct differences in the serum proteins of the inflammation and anti-tumor response spectrum of VLD patients. 

Many questions remain to be addressed about VLD in melanoma. The aim of this prospective study was to explore the proteomic immune signature of VLD, to characterize its clinical presentation and to investigate the potential association with the clinical response under CPI. 

## 2. Materials and Methods

This monocentric, prospective observational cohort study was performed at the Department of Dermatology of the University Hospital Zurich, Switzerland. Adult melanoma patients (≥18 years) treated with CPI who developed VLD were included between 05/2018 and 08/2020. 

Our study population consisted of four different treatment cohorts receiving the following immunotherapies: a combination of ipilimumab and nivolumab, anti-PD-1 monotherapy with pembrolizumab or nivolumab and clinical trials. All patients received immunotherapies according to standard dose and schedule. Prerequisites for inclusion were advanced melanoma and CPI-associated VLD. Exclusion criteria for the subanalysis regarding responsiveness included adjuvant immunotherapy. One additional patient was eliminated from the analysis due to unclear data.

### 2.1. Skin Assessments

At two to three-week interval visits, patients were screened for VLD manifestations by a dermatologist. VLD was defined as the appearance of depigmented skin lesions and categorized as symmetrical vs. asymmetrical, small macules (≤1 cm) vs. large patches (>1 cm), with a universal, polymorphous, or confluent distribution. Macules, as well as patches, are flat, non-elevated lesions which are only distinguished by their size. The clinical course of VLD, including the body surface area (BSA) measurement, was systematically recorded by photography of affected areas. In addition, Wood’s light examination was used to confirm the diagnosis. The photographs were analysed as to the extent of VLD (affected BSA) and localization (head, neck, trunk, upper and lower extremities, décolleté, upper back) by two independent blinded investigators. In addition to VLD, all irAEs were documented according to the current standards (Common Terminology Criteria for Adverse Events CTCAE v4.03). Regular stagings with PET-CT were performed every three months using RECIST criteria (version 1.1) [21] according to the eighth edition of the American Joint Committee on Cancer melanoma staging and classification [22].

The study was conducted according to the Helsinki Declaration and was approved by the local ethics committee (EK2019-01825). Data protection, according to the EU and Swiss standards, was guaranteed and enforced for all study patients.

### 2.2. OLINK High-Throughput Proteomics

For serological protein analysis, OLINK assays were performed using two panels: Inflammation and Immune Response (Olink, Uppsala, Sweden). The proximity extension assay (PEA) was performed according to manufacturer’s instructions. Briefly, protein-specific antibodies tagged with DNA sequences were used, followed by a PCR where matched DNA reporter pairs served as the template. In the last step, a Biomark HD system (Fluidigm, San Francisco, CA, USA) device was used for microfluidic qPCR and therefore signal quantification. Data were analysed using R 4.1.2 (R Core Team, R Foundation for Statistical Computing organization, Vienna, Austria) and the package OlinkAnalyze 3.1.0 (Olink Proteomics Data Science Team, Uppsala, Sweden) and OlinkR1.0.1 (Olink Proteomics Data Science Team, Uppsala, Sweden) as well as for visualisation ComplexHeatmap 2.10.0 [23] and EnhancedVolcano 1.12.0 [24]. 

### 2.3. Histopathological Examination, Immunohistochemistry Staining

In all patients, the diagnosis of VLD was histopathologically confirmed by a board-certified dermatopathologist. Loss of melanin pigmentation and melanocytes were evaluated in nine patients using Masson Fontana stain and immunohistochemical stainings (Melan-A, Sox-10 and Tyrosinase). 

### 2.4. Data Analysis and Statistics 

Descriptive statistics and visualizations were elaborated in RStudio. For further statistical evaluations, such as the examination of variance homogeneity and significance calculations, SPSS was used. For the variance homogeneity examination, Levene’s test was applied, a *t*-test for independent groups. In case of confirmed variance homogeneity and to determine the significance, a 2-sided *t*-test was performed. The significance was calculated with a confidence interval of 95%. For the pattern analysis, an Anova test was applied. For survival analyses, a Schoenfeld’s test was conducted to check the proportional hazards assumption, a Kaplan-Meier estimator used to plot survival curve and the log-rank test applied to assess if the curves were significantly different. 

Analyses of the Olink protein data were processed applying the “OlinkAnalyze” R package provided by Olink (Uppsala, Sweden). For the comparison of two groups, a *t*-test was used, whilst multiple groups were analysed by applying an ANOVA F-test. The Benjamini–Hochberg method (“fdr”) was used for the multiple testing correction. Heatmaps and volcano plots were created using the limma function of the OlinkR package as well as the EnhancedVolcano and ComplexHeatmap packages. 

## 3. Results

### 3.1. Demographics and Patient Characteristics

A total of 28 VLD patients were included with a mean observation period of 26.8 months ± 19.46 months. The mean age of patients at VLD diagnosis was 66.04 ± 13.50 years. 68% of the VLD patients were male, and the male-to-female ratio was approximately 2:1. At the time of diagnosis, male patients were slightly older (66.84 ± 13.01 years) in comparison to female patients (64.33 ± 15.15 years). All patients suffered from stage IIIB-IV melanoma at the time of VLD diagnosis (details are shown in Table 1). Most patients (n = 22) were under conventional immunotherapies (ipilimumab, nivolumab, pembrolizumab) at the time of VLD diagnosis. Six patients were part of clinical trials (receiving an anti-PD-1 antibody such as spartalizumab, and/or other additional agents described in Table 1 legends). 

### 3.2. The Distinct Molecular Signature of VLD

The occurrence of VLD in melanoma patients under immunotherapy has been described as one of the most closely associated irAEs with improved clinical outcome [9]. We therefore hypothesized that among melanoma patients receiving immunotherapy, those with VLD might exhibit a distinct molecular signature compared to those without VLD, either with or without other irAEs. We performed a high-throughput proteomics assay OLINK with serum samples from 26 of our VLD patients as well as seven age- and sex-matched CPI-treated non-VLD (nVLD) patients (Appendix A), half of them with and half of them without other irAEs. A principal component analysis showed a separation and a more diverse picture in the VLD cohort (Figure 1A). Of note, several proinflammatory chemokines were reduced in VLD patients, e.g., CXCL5 and CXCL6, whereas ITGA11 was increased (Figure 1B–D). Subgrouping the nVLD patients into those with and without irAEs revealed differences especially regarding beta-NGF and TGF-α (Figure 1E). 

### 3.3. Clinical and Histopathological Characterization of VLD

We wanted to provide a thorough characterization of VLD. VLD lesions showed a “milky” white appearance under Wood’s light examination (as in classical vitiligo). We systemically assessed lesions with regard to their morphology and distribution. We observed a significantly higher percentage of patients with a symmetrical rather than with an asymmetrical distribution of VLD lesions (*p* ≤ 0.001) (Figure 2A,B vs. Figure 2C and Figure 3). Slightly more patients presented with small freckle-like macules than with patches (*p* = 0.19) (Figure 2A, Table 2, Figure 3). In all immunotherapy subgroups, the VLD lesions predominantly occurred in sun-exposed areas such as the face, the upper and lower extremities and the trunk involving the cleavage/upper back (Figure 2E). All patients developed VLD lesions in at least one sun-exposed area. The median affected BSA was 12.5%. Regarding possible similarities with distribution patterns observed in classical vitiligo, we found that mechanically stressed areas (e.g., eyelids, perioral area, dorsal hands, and fingertips), which are subject to the Koebner phenomenon, were also predominantly affected in VLD. In 57% of the patients, at least one of the mechanically stressed areas showed depigmentation. The most frequently affected areas were the fingertips, joints of the hands and the periocular region (Figure 2D). Three of the patients with asymmetrical VLD had skin lesions that had developed a halo phenomenon around cutaneous metastases (Figure 2F).

Histopathological assessment of nine VLD patients demonstrated a complete loss of melanin pigmentation (demonstrated with Masson-Fontana stain) in 9/9 cases. Melanocytes (stained with anti-Melan-A, -SOX-10, and -tyrosinase antibodies) were completely absent in 5/9 (55.5%) and significantly reduced in number in 4/9 (44.4%) specimens. 

### 3.4. Overall Responsiveness to Immunotherapy

Next, we aimed to investigate the responsiveness of VLD patients to immunotherapy. Our patients had received a combination of ipilimumab and nivolumab (9/28, 36%), anti-PD-1 monotherapy with pembrolizumab (10/28, 36%) or nivolumab (3/11%), other immunotherapy-based clinical trial medications (6/28, 21%) or anti-CTLA-4 monotherapy prior to VLD onset. Excluding the adjuvant patients and one patient with unclear data, the complete remission rate was 24% (n = 6), partial remission rate was 28% (n = 7), stable-disease rate was 8% (n = 2) and progressive disease rate was 40% (n = 10) at VLD-onset. Three years following VLD onset, the probability for progression-free survival was 60.4% (95% CI: 43–84%) (Figure 4A) and the overall survival was 73.7% (95% confidence interval 57.5 to 94.6%) (Figure 4B). 

The mean overall time lapse of VLD onset after CPI start was 7.2 (±3.7) months; the median time lapse was 6.5 months. We compared patients who had received a targeted therapy beforehand (n = 4) to patients who had not (n = 20). Four patients were excluded from this analysis due to missing data. Patients who had previously received targeted therapy developed VLD significantly later (6.5–7.2 months) than patients with no prior treatment (*p* ≤ 0.001). The median time until VLD onset in the non-pre-treated group was 6.25 months (range 1–13 months) and the median average time in the pre-treated group was 12.5 months (range 9.5–16 months) (Figure 5A).

### 3.5. Parameters Associated with a Better Outcome 

We were interested in whether we might identify particular clinical or molecular parameters associated with a better treatment response among our VLD patients. Regarding the clinical VLD pattern (asymmetrical/symmetrical, macular/patches, with/without Koebner), we did not find any differences between them in terms of PFS or OS (Table 3 and Table 4. We also examined whether there was a difference in the time of VLD onset between patients showing an objective radiological response and patients with stable or progressive disease. 

The responders group (partial remission or complete remission at VLD onset; n = 13) had a mean time of 6.92 (±3.16) months and median time of 6.5 months until VLD development, vs. a mean time of 7.58 (±4.28) months and median time of 7.5 months until VLD onset in the non-responders group (n = 12). The observed differences were not significantly different (*p* = 0.81) (Figure 4B).

When exploring the potential association with biological parameters, we first looked into the presence of a BRAF-V600E mutation. The latter was not associated with a different outcome (*p* = 0.49). A normal serum lactate dehydrogenase (LDH) was associated with improved outcome (Appendix A). 

### 3.6. Molecular Immune Profiles Associated with a Better Outcome 

Finally, we wanted to address whether VLD patients with a better outcome might exhibit a particular molecular immune response profile. We hypothesized that the different profiles we saw in the PCA (Figure 1B) were two different subtypes apparent in the VLD group with either high or low inflammation signatures. There was, however, no appropriate clinical correlate to this. Our OLINK analyses showed overlapping clusters in the relevant PCA analyses (Figure 6A). When investigating the variety of proteins of the inflammation and immune response panel, we identified different inflammation signatures comparing responders vs. non-responders among VLD patients (Figure 6B). Whilst several proinflammatory receptors (e.g., TREM1, CLEC4A, DCTN1) and kinases (PRKCQ) were upregulated in the responder group, the costimulatory molecule LAG3, IFN-γ and IL-18R1 were significantly decreased (Figure 6C–E). In the survival analysis, we identified ITGA11 as the most closely associated protein with an improved overall survival; the difference however was not significant after multiple test correction (*p* = 0.147) (Appendix A).

## 4. Discussion

In this study, we explored the systemic immune signatures and characterized clinical patterns of VLD and evaluated whether they are associated with different treatment outcomes in melanoma patients receiving CPI [9].

Our proteomic analysis revealed a distinct immune signature of VLD in comparison to CPI recipients without VLD, both with/without other irAEs. Of note, several proinflammatory proteins, such as CXCL5, CCL11 and TNFSF14, were significantly reduced in VLD. There were no clear differences investigating nVLD with and without other irAEs; however, the sample size was limited regarding a direct comparison. Based on these data, one may hypothesize that a specific constellation of cytokines may induce or indicate VLD. Of note, ITGA11 especially was increased in VLD patients. ITGA11 belongs to the group of integrins which are transmembrane proteins that mediate cell adhesion to the extracellular matrix [25]. They play an important role in the regulation process of cell proliferation, migration, differentiation, tumor invasion and metastasis [26]. Integrin α11 (ITGA11) is a known receptor for collagen. The role of this protein has been shown, especially in lung cancer and gastric cancer tissue, as being associated with tumor progression [25,27]. The extent to which it is involved in the induction of VLD needs further investigation.

Clinically, VLD preferentially develops in sun-exposed areas. The sun-exposed area-distribution seems to be a VLD-specific phenomenon which is not commonly observed in non-immunotherapy-associated vitiligo [11,20,26]. Our VLD distribution findings are in line with a cross-sectional study with 85 VLD patients [28]. Due to the partial overlap of mechanically stressed and UV-exposed areas, we cannot exclude an overlap bias in our observations. In mechanically stressed (Koebner) areas, depigmented lesions were detected in more than 50% of the VLD patients, which is similar to what has been described in non-immunotherapy-associated vitiligo (21–62%) [29]. The biological mechanism underlying the preferential development of VLD lesions in UV-exposed areas has not been elucidated. Our observations suggest that the innate UV-sensing danger-signaling cascade triggering T-cell activation and, ultimately, melanocyte destruction, may be altered in melanoma VLD patients. The mechanisms underlying this phenomenon and how it relates to the antitumoral immune response in melanoma need to be further explored. A comparison of the VLD patterns between different Fitzpatrick phototypes might be interesting to assess the protective effect of melanin. Another clinically interesting finding was the initial occurrence of VLD after CPI [11,13,27]. When analysing VLD onset, we found that patients who were pre-treated with targeted therapy had a significantly later onset: the median time to VLD onset was 12.5 months in the kinase-inhibitor pre-treated group vs. 6.25 months in the group without prior targeted therapy. Targeted therapy thus seems to impact the (antitumoral) immune response induced by CI treatment. This could occur directly via their effect on T lymphocytes, as evidenced, e.g., by an increased CD8+ component in tumor-infiltrating lymphocytes [30], or indirectly by altering the tumor microenvironment and cytokine milieu [16,31]. We observed that a later VLD onset is not associated with a distinct therapeutic outcome. However, this argues against a pathomechanistic significance of this finding with regard to the antitumoral response.

Histopathological evaluation revealed findings compatible with classical spontaneously occurring vitiligo [32,33], i.e., a partial or complete loss of melanocytes in all VLD specimens. Identic histopathological findings in VLD have been described by Ramodetta et al. [34]. 

The clinical pattern analysis of VLD revealed a predominance of symmetrically distributed macular depigmented lesions, whereas patches and an asymmetrical pattern were less prevalent. Based on our findings, we hypothesized that this particular VLD pattern may be associated with a distinct underlying T-cell response and thus a different clinical outcome. This theory was not confirmed by our analysis of outcome in terms of OS and PFS, which was not different in VLD with vs. without a macular pattern. We did, however, find that biochemical factors (a low baseline LDH) correlated with a significantly better outcome (PFS and OS). Our findings confirm the association of an LDH above the upper normal limit with an inferior outcome, which has previously been observed in non-VLD ICP cohorts [1,35]. The answer to the question of whether VLD patients without additional irAEs have a better treatment response in comparison to VLD patients with other irAEs was negative in our study population. Meanwhile, previous studies have indicated that melanoma patients who develop VLD have a better therapeutic response to immunotherapy [10,12,27,33,34]. A higher percentage of BSA affected by VLD and trend of expansion have previously been suggested as predictive markers for prolonged survival under CPI [36]. This was not the case in our cohort. It remains to be elucidated whether VLD better reflects the clinical correlate of a CPI-induced antitumoral immune response in melanoma patients than other irAEs. Regarding survival in a multi-center meta-study by Guida et al. [37], which included 148 VLD patients treated with anti-PD1 in combination with/without anti-CTLA-4, the 36-month PFS of 52% is below our PFS-result (60.4%, n = 26). On the other hand, Guida et al. reported a longer three-year OS with 82% than was shown in our cohort (73.7%, n = 26). One factor that may have contributed to an inferior OS in our cohort could be the serum LDH levels, which were elevated in 54% of our patients compared to only 26% of patients in the meta-analysis. In mixed (VLD and nVLD–developing) treatment-naïve melanoma patients treated with pembrolizumab, nivolumab and nivolumab in combination with ipilimumab, the three-year OSs were 51%, 52% and 58%, respectively [38,39]. Although an inter-study comparison has limited power, our VLD only cohort showed somewhat superior values, with a 73.7% three-year OS. Considering that our cohort included 30.7% (8/26) pre-treated patients with locally unresectable or metastatic melanoma, the superior three-year OS emphasizes the positive prognostic value of VLD [40].

Interestingly, responders vs. non-responders within the VLD group showed much more heterogeneity in their proteomic immune profiles than those without VLD. Responders showed several increased proteins, such as EDAR and PLXNA4, whereas LAG3 and other proteins were reduced. LAG3 is an inhibitory molecule that is targeted by novel treatments (relatlimab) [41]. The downregulation in responders may lead to a reduced inhibition of T cells and, therefore, a better antitumor immune response. Of note, a proapoptotic function of EDAR has been described [42]. Vial et al. hypothesize that EDAR may constrain melanoma progression. AXIN1 was increased in responders as well. Biechele et al. described AXIN1 as a potential mediator of apoptosis [43]. PLXNA4 was already proposed by Celus et al. as a potential biomarker for therapy response [44], which is in line with our results. Further investigations are needed to analyse their potential role as predictive markers for therapy response. 

We compared our data to the TCGA database and no relevant information regarding the according genes to the significant up- and downregulated proteins was found. 

While this study includes a large VLD cohort in a single center, caution is warranted when interpreting our results due to several limitations. For the targeted high-throughput proteomics (Olink) experiments, the sample size of the control group was quite small (n = 7), since in the current project we focused on the characterization and stratification within the VLD group itself. Our presented findings need to be confirmed in larger cohorts, which should include larger control groups without VLD, but only with other irAEs and without any irAEs. The heterogeneity of our VLD patient population could be an additional source of bias and underlines the need for further sub-analyses according to different tumor stages, CPI combinations and pre-treatments. Future studies should seek to investigate the different pathomechanism underlying the development of VLD compared to classical vitiligo to better characterize the T-cell-mediated mechanism in both conditions.

## 5. Conclusions

Our results highlight the distinct proteomic immune signatures, such as an upregulation of EDAR and downregulation of LAG3, in responders among VLD melanoma patients in comparison to non-responders. The clinical characterization of VLD in melanoma patients has revealed a predominance of a symmetrical, small macular pattern of depigmentation in UV-exposed areas without a specific pattern showing an advantage in tumor response or survival. The significantly longer time to VLD onset in the kinase-inhibitor pre-treated group is an additional interesting new finding which gives us clues as to the immunological factors contributing to VLD. Futures studies with larger sample sizes should investigate how our findings relate to the antitumoral immune response and what we could extrapolate from this to the clinics.

## Figures and Tables

**Figure 1 cancers-14-04576-f001:**
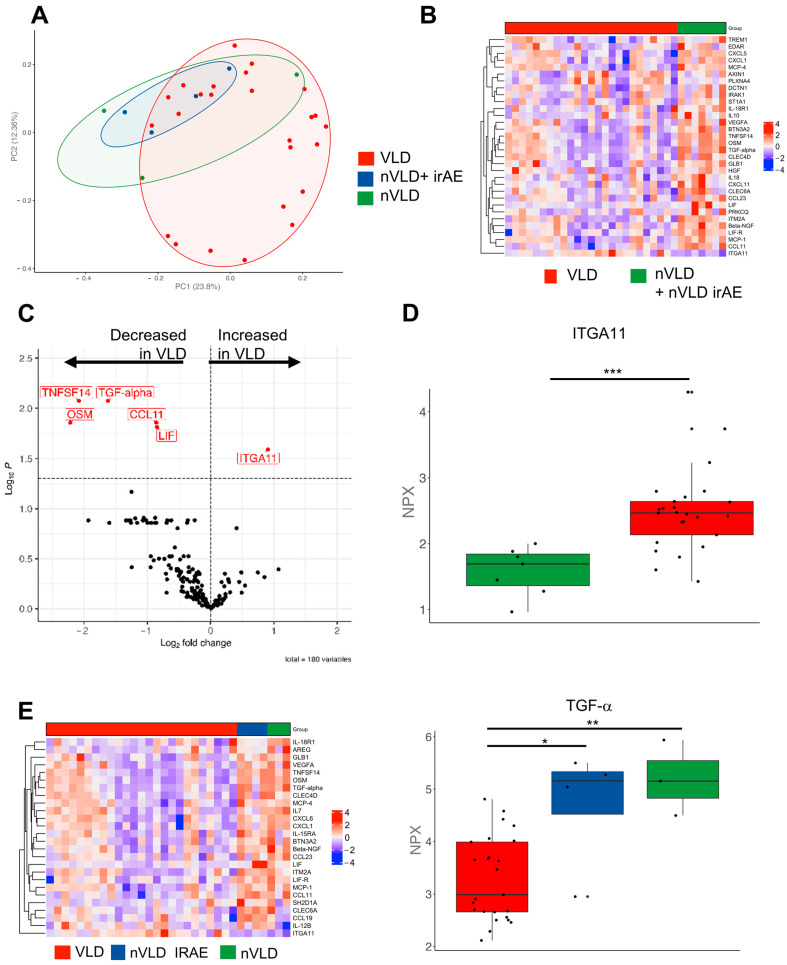
Serum protein analysis investigating differences between VLD and non-VLD (nVLD) patients. (**A**) Principal component analysis comparing VLD with nVLD patients (with nVLD irAE and without irAE). (**B**) Heatmap depicting VLD and nVLD patients. (**C**) Volcano plot showing several decreased proteins in VLD patients compared to nVLD individuals. (**D**) Boxplot showing the normalized protein expression (NPX) of ITGA11 as the most upregulated protein in VLD patients (***, *p* < 0.001). (**E**) Heatmap showing the significant different proteins (ANOVA) comparing all three groups (**, *p* < 0.01; *, *p* < 0.05). TGF-α showed the highest value in nVLD without irAE and the lowest in VLD patients. ITGA11: Integrin alpha-11, TGF-α: Transforming Growth Factor alpha.

**Figure 2 cancers-14-04576-f002:**
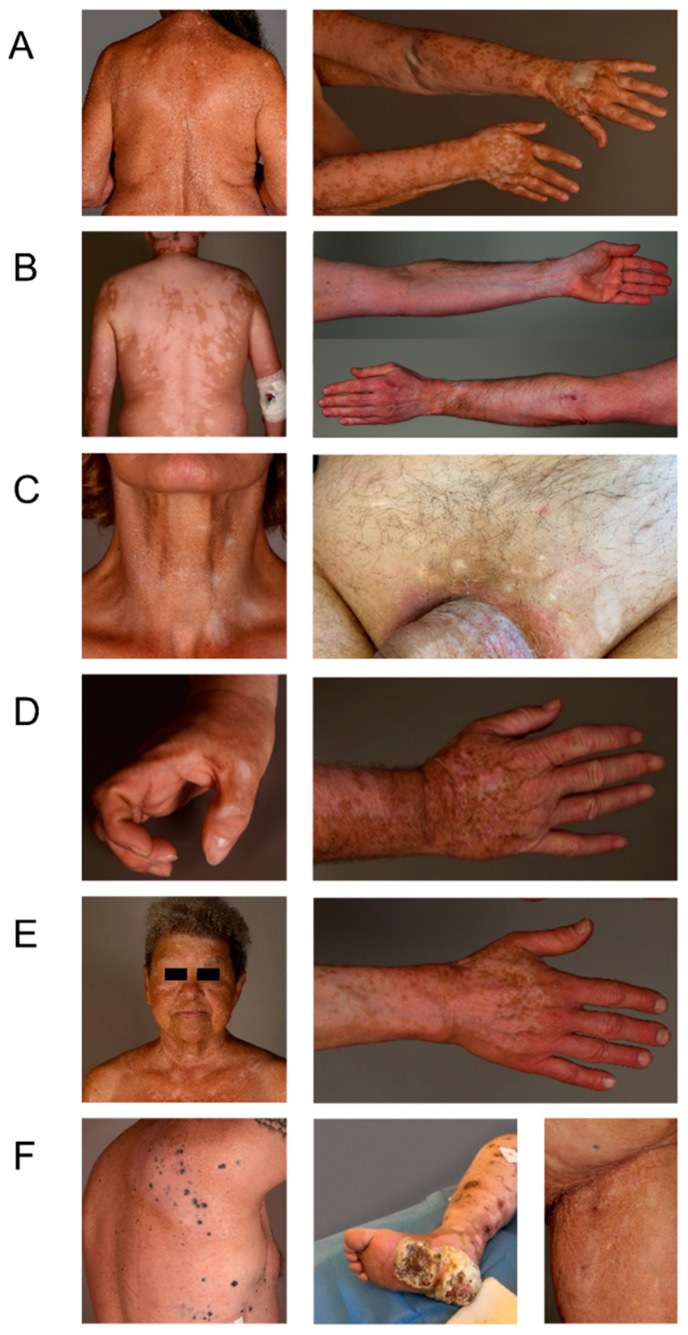
Different VLD patterns. (**A**) Small macule pattern, symmetrical distribution, partly confluent on the arms, representing the most prevalent pattern. (**B**) Large patch pattern, symmetrical distribution, confluent on the back, arms and legs. (**C**) Asymmetrical distribution on the neck and on the groin area. (**D**) Areas with Koebner’s phenomenon. (**E**) Sun-exposed areas without Koebner’s phenomenon. (**F**) Halo phenomenon around cutaneous metastases, asymmetrical pattern.

**Figure 3 cancers-14-04576-f003:**
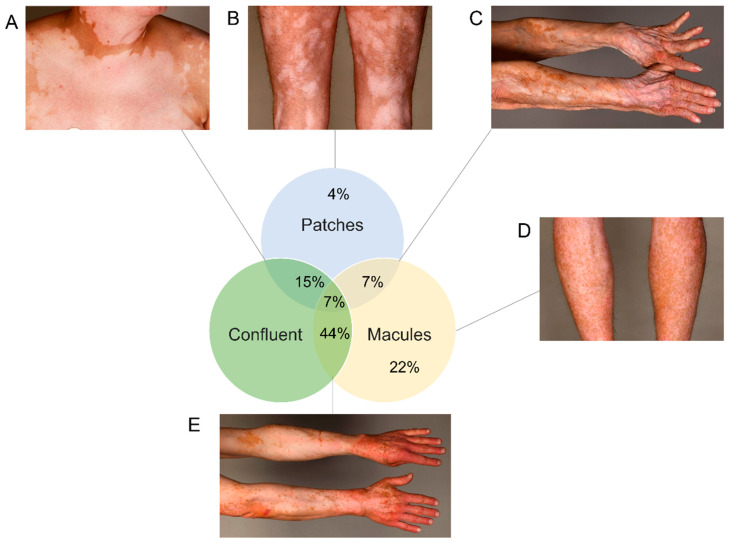
Different patterns in absolute frequencies with examples. (**A**) Confluent big patches. (**B**) Large patch-pattern non-confluent. (**C**) Patches and macules, confluent pattern. (**D**) Small macules, non-confluent. (**E**) Confluent macules (most frequent pattern).

**Figure 4 cancers-14-04576-f004:**
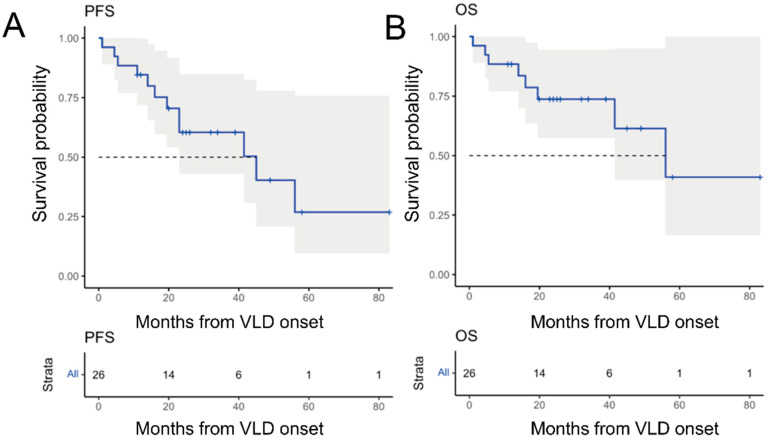
Kaplan–Meier survival analysis and prediction in 26 patients developing VLD after receiving immunotherapy for metastatic melanoma. (**A**) Progression-free survival of all VLD patients with a 3-year PFS of 60.4% (95% confidence interval 43–84%). (**B**) OS of all VLD patients with a 3-year OS of 73.7% (95% confidence interval 57.5–94.6%).

**Figure 5 cancers-14-04576-f005:**
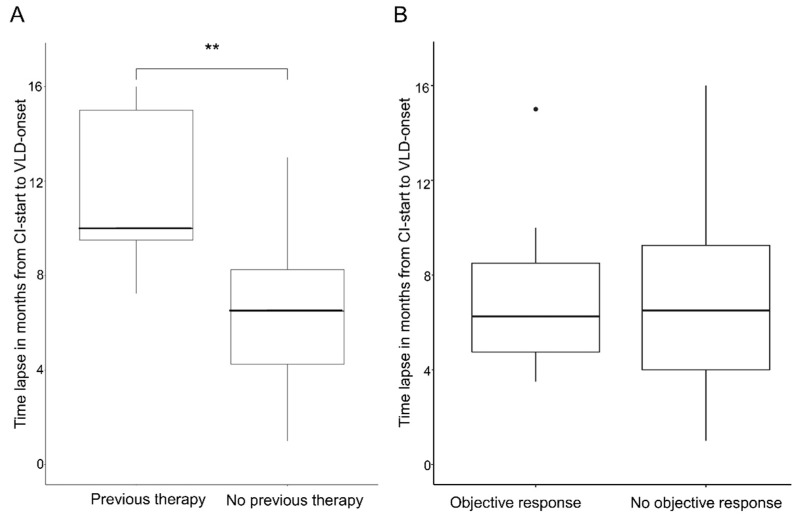
Time lapse between start of CPI and VLD onset. (**A**) Box plot showing a significant difference in the time lapse of VLD onset between patients with previous targeted therapy vs. no previous targeted therapy (** *p* < 0.001). (**B**) Box plot showing no significant time lapse of VLD onset of therapy responders vs. non-responders.

**Figure 6 cancers-14-04576-f006:**
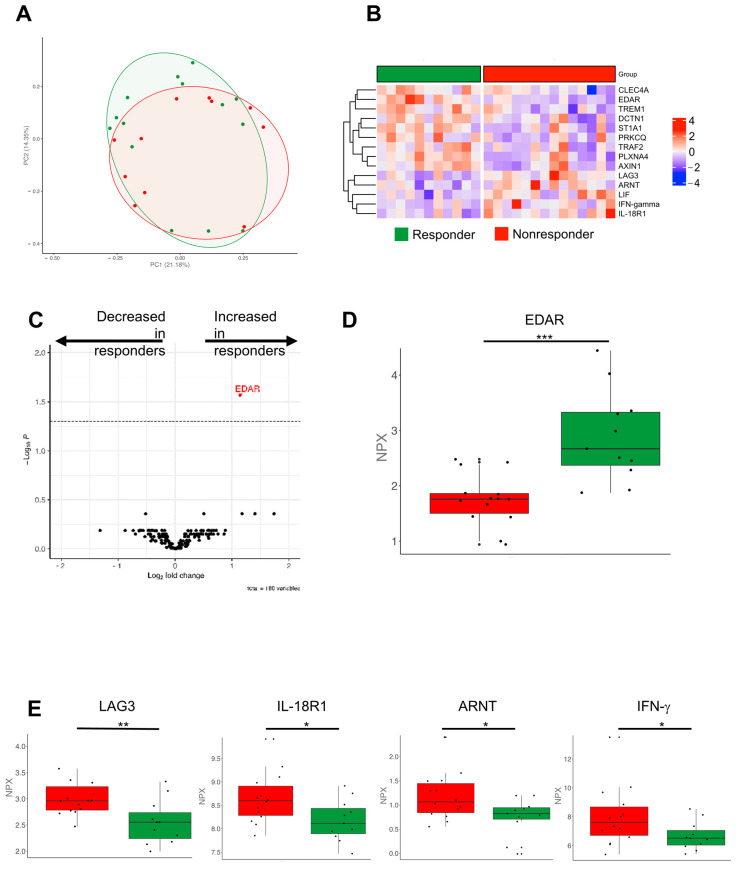
Olink protein analyses comparing responder and non-responder VLD patients. (**A**) Principal component analysis showing an overlap of responder (green) and non-responder (red). (**B**) Heatmap of the significant different proteins compared by applying *t*-test. Z-score scaling across the rows are depicted. (**C**) Heatmap showing the log2 fold changes and *p*-values comparing responders and non-responders. (**D**) Boxplot showing the normalized protein expression of EDAR as one of the upregulated proteins in responders with the highest *p*-value (***, *p* < 0.001). (**E**) LAG3, IL-18R1, ARNT, IFN-γ showed significant increase in the non-responder group applying *t*-test (**, *p* < 0.01; *, *p* < 0.05). ARNT: Aryl Hydrocarbon Receptor Nuclear Translocator, EDAR: Ectodysplasin A Receptor, IFN-γ: Interferon gamma.

**Table 1 cancers-14-04576-t001:** Baseline characteristics.

Clinical and Disease Features of Patients with VLD	
Characteristics (n = 28 patients)	
Sex, n (%)	
Female	9 (32)
Male	19 (68)
Checkpoint inhibitor, n (%)	
Nivolumab	3 (11)
Pembrolizumab	10 (36)
Ipilimumab/Nivolumab	9 (32)
Clinical Studies *	6 (21)
Type of melanoma, n (%)	
ALM	1 (4)
Nodular	7 (25)
Not clear	4 (14)
Sinunasal	3 (11)
SSM	4 (14)
Unknown primary	5 (18)
Uveal	3 (11)
NA	1 (4)
Previous targeted therapy	
Yes	5 (18)
No	22 (79)
NA	1 (4)
BRAF mutated	
Yes	9 (32)
No	18 (64)
NA	1 (4)
LDH elevated	
Yes	15(54)
No	13 (46)
Tumor stage at melanoma diagnosis, n (%)	
I	1 (4)
I-II (uveal)	3 (11)
IIB	2 (7)
IIC	1 (4)
IIIB	8 (29)
IIIC	6 (21)
IV	6 (21)
NA	1 (4)
Tumor stage at VLD diagnosis, n (%)	
IIIB	1 (4)
IIIC	3 (11)
IV	24 (86)
Line of immunotherapy before VLD onset, n (%)	
Adjuvant	2 (7)
1st line	20 (71)
2nd line	5 (18)
3rd line	1 (4)
Line of systemic therapy in general before VLD onset, n (%)	
1st line	20 (71)
2nd line	5 (18)
3rd line	2 (7)
4th line	1 (4)

ALM, acrolentiginous melanoma; SSM, superficial spreading melanoma; NA, not available. * Other medication given in the clinical trials were ribociclib (CDK4-6 inhibitor), lenvatinib (multi-kinase-inhibitor), MCS 110 (humanized macrophage colony stimulating factor), epacadostat (inhibitor of indoleamine 2,3-dioxygenase-1 (IDO1)) and tebentafusp (T-cell redirecting agent), the latter having been recently approved for monotherapy in metastatic uveal melanoma.

**Table 2 cancers-14-04576-t002:** Distribution of VLD skin lesions and clinical features: Analysis of the distribution and appearance of the depigmented lesions.

Clinical and Disease Features of Patients with VLD	
Characteristics (n = 28 patients)	
Distribution pattern, n (%)	
Symmetric	20 (71)
Asymmetric	5 (18)
NA	3 (11)
Face, n (%)	
Yes	15 (54)
No	5 (18)
NA	8 (29)
Scalp, n (%)	
Yes	8 (29)
No	7 (25)
NA	13 (46)
Hair, n (%)	
Yes	8 (29)
No	8 (29)
NA	12 (43)
Acral, n (%)	
Yes	20 (71)
No	3 (11)
NA	5 (18)
Upper extremities, n (%)	
Yes	23 (82)
No	2 (7)
NA	3 (11)
Lower extremities, n (%)	
Yes	11 (39)
No	6 (21)
NA	11 (39)
Genital area, n (%)	
Yes	3 (11)
No	4 (14)
NA	21 (75)
Oral mucosa, n (%)	
Yes	3 (11)
No	7 (25)
NA	18 (64)
Neck trunk sun-exposed, n (%)	
Yes	21 (75)
No	3 (11)
NA	4 (14)
Lower trunk, n (%)	
Yes	13 (46)
No	7 (25)
NA	8 (29)
Trunk general, n (%)	
Yes	20 (71)
No	4 (14)
NA	4 (14)
Koebner areas, n (%)	
Yes	16 (57)
No	11 (39)
NA	1 (4)

VLD: vitiligo-like depigmentation. NA: not available.

**Table 3 cancers-14-04576-t003:** Univariable and multivariable Cox regression analyses of the progression-free survival.

Univariable and Multivariable Cox Regression Analyses of PFS (n = 26)					
Variable	Hazard Ratio (95% CI)	*p*-Value	Variable	Hazard Ratio (95% CI)	*p*-Value
PFS univariate analysis			PFS multivariate analysis		
LDH elevated		**0.03**	LDH elevated		**0.02**
Yes	1		Yes	1	
No	0.18 (0.04–0.83)		No	0.12 (0.021–0.73)	
Dist symmetric		0.94	Dist symmetric		0.80
Yes	1		Yes	1	
No	0.95 (0.24–3.81)		No	0.79 (0.15–4.32)	
BRAF mutated		0.46	BRAF mutated		0.49
Yes	1		Yes	1	
No	1.65 (0.44–6.16)		No	1.84 (0.32–10.49)	
Koebner		0.85	Koebner		0.61
Yes			Yes		
No	1.12 (0.35–3.58)		No	0.62 (0.097–3.94)	
Patches		0.56	Patches		0.81
Yes	1		Yes	1	
No	0.71 (0.22–2.25)		No	0.77 (0.09–6.47)	
Macules		0.49	Macules		0.27
Yes	1		Yes	1	
No	1.52 (0.45–5.17)		No	0.27 (0.025–2.80)	
Confluent		0.52			
Yes	1				
No	1.48 (0.45–4.78)				

Bold entries are statistically significant values. PFS, progression free-survival; CI, confidence interval; Dist, distribution; LDH, lactate dehydrogenase; OS, overall survival; VLD, vitiligo-like depigmentation.

**Table 4 cancers-14-04576-t004:** Univariable and multivariable Cox regression analyses of the overall survival.

Univariable and Multivariable Cox Regression Analyses of OS (n = 26)					
Variable	Hazard Ratio (95% CI)	*p*-Value	Variable	Hazard Ratio (95% CI)	*p*-Value
OS univariate analysis			OS multivariate analysis		
LDH elevated		0.06	LDH elevated		0.05
Yes	1		Yes	1	
No	0.13 (0.016–1.06)		No	0.10 (0.01–1.04)	
Distribution symmetric		0.41	Distribution symmetric		0.44
Yes	1		Yes	1	
No	1.96 (0.40–9.53)		No	2.33 (0.27–19.8)	
BRAF mutated		0.55	BRAF mutated		0.32
Yes	1		Yes	1	
No	1.63 (0.32–8.20)		No	4.02 (0.27–60.4)	
Koebner		0.9	Koebner		0.26
Yes			Yes		
No	0.9 (0.21–3.8)		No	0.32 (0.04–2.33)	
Patches		0.26	Macules		0.17
Yes	1		Yes	1	
No	0.43 (0.10–1.85)		No	0.14 (0.001–2.39)	
Macules		0.37			
Yes	1				
No	1.93 (0.45–8.19)				
Confluent		0.34			
Yes	1				
No	1.99 (0.48–8.30)				

PFS, progression free-survival; CI, confidence interval; Dist, distribution; LDH, lactate dehydrogenase; OS, overall survival; VLD, vitiligo-like depigmentation.

## Data Availability

Anonymized data collection tables can be requested from the corresponding authors.

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
