# Peer review of "Clinical Presentation and Prognostic Features in Patients with Immunotherapy-Induced Vitiligo-like Depigmentation: A Monocentric Prospective Observational Study"

_cancers, 2022, doi:10.3390/cancers14194576_

Round 1

Reviewer 1 Report

Extensive and good report.

It would be better to define more clearly the manifestation of vitiligo-like depigmentation.

1. Specify more clearly the clinical differences between VDL and "ordinal vitiligo". Why does the VLD show depigmented "plaque" lesions? Vitiligo always reveal macular lesions.

2. Why VLD is more commonly seen on sun-exposed area? Is this difference related to the fact that depigmentation is more "preferentially" seen in the sun/UV-exposed pigmentated skin lesion in non-oriental white-skinned people?

3.  Are clinical evaluations of depigmentation carried out under Wood lamp examination? If they were done so, does VLD lesion show the "milky" white depigmentation as ordinal vitiligo shows characteristically under Wood light?

4. Are VLD lesions show any "remaining" melanocytes by immuno-histochemical  markers and Fonta Masson staining? I f done so, what are the differences between VLD and vitiligo lesions?

Reviewer 2 Report

Hermann et al. elucidate a proteomic signature associated with Vitiligo-like depigmentation (VLD), an adverse event associated with CPI treatment in a proportion of melanoma patients. The OLINK technology is well suited for exploring a selected panel of markers (which may comprise drivers or passenger events) linked with the onset/progression of VLD. The authors have carefully tempered their conclusions to not overinterpret. They have also included a neat graphical abstract that appropriately conveys the concept and findings, which is appreciable.

It is of overwhelming interest to elucidate the broader proteome and possibly PTM (especially phosphorylation) signatures associated with VLD using unbiased screening methods such as by mass spectrometry. It is hoped that the authors would consider pursuing such efforts in the future as this will more likely lead to functionally important cues into the pathophysiology of VLD.   

- A major concern with this study relates to the poor sample size of the nVLD cohort (7, of which only 3 are nVLD without any IRAE) which makes the study absurdly disbalanced. This potentially skews the dataset and creates doubts on value of the signature reported. The authors need to accept this in the manuscript and provide a strong supporting justification.

-The authors speak to elevated LDH being associated with poor survival. This is not surprising since serum LDH is very strong independent prognostic factors in melanoma that has been reported by several groups. So there it is strange that there is particular emphasis over this finding in the present study.

- There does not seem to be access to supplementary tables. It is strongly suggested that the authors include a supplementary table, if not already done so, which comprehensively describes the OLINK data. This table should list the proteins, fold-change, p-value, adjusted p-value and actual normalized protein intensities used to calculate the fold-change.

- Do any of the top differentially expressed proteins associate with poor prognosis? This is more interesting and important as a result than that shown for LDH. The authors are encouraged to discuss appropriately the involvement of these proteins in other types of cancers, as reported in the literature.

- The authors must also consider exploring TCGA database for expression and prognosis of their differentially expressed proteins. Several user-friendly web-based tools exist for his purpose.      

- There is some confusion over how the authors distinguish IRAE and VLD cases. VLD is a type of IRAE (from line 68) but the authors seem to talk about IRAE elsewhere in the manuscript as though it belongs to a completely different class.

- Line 166: please be consistent; is it “CI-treated” or “CPI-treated” ?

- Line 117: replace “Shortly” with “Briefly”

- Be consistent; irAE vs IRAE. Both are used throughout the manuscript.

Minor grammatical/ language issues and typos persist throughout the manuscript which must be carefully identified and corrected. E.g. line 288 “according PCA analysis”

Round 2

Reviewer 2 Report

The authors provide reasonable justification and it is hoped that a follow-up of this work would be performed via deep unbiased proteomics.